# AMORTIZED VARIATIONAL COMPRESSIVE SENSING

**Aditya Grover & Stefano Ermon**
Computer Science Department
Stanford University
{adityag,ermon}@cs.stanford.edu

## ABSTRACT

The goal of statistical compressive sensing is to efficiently acquire and reconstruct high-dimensional signals with much fewer measurements, given access to a finite set of training signals from the underlying domain being sensed. We present a novel algorithmic framework based on autoencoders that jointly *learns* the acquisition (a.k.a. encoding) and recovery (a.k.a. decoding) functions while implicitly modeling domain structure. Our learning objective maximizes a variational lower bound to the mutual information between the signal and the measurements. Empirically, we show $20 - 46\%$ improvement in reconstruction accuracies over competing approaches on the MNIST dataset for the same number of measurements.

## 1 INTRODUCTION

Efficient acquisition and recovery of high-dimensional signals is important for many machine learning applications such as magnetic resonance imaging and remote sensing (Lustig et al., 2007; Herman & Strohmer, 2009). In compressive sensing, we formalize this problem as solving a system of linear equations for an $n$-dimensional signal $x^* \in \mathbb{R}^n$ using $m$ measurements $y \in \mathbb{R}^m$ related as:

$$y = Ax^* + \epsilon \tag{1}$$

where the linear map $A \in \mathbb{R}^{m \times n}$ is referred to as the measurement matrix and $\epsilon \in \mathbb{R}^m$ is the measurement noise. If $m << n$, then the system is underdetermined and additional assumptions are required to guarantee a unique recovery. The celebrated results in the compressive sensing literature posit that a $k$-sparse signal can be recovered with high probability using only $O(k \log \frac{n}{k})$ measurements acquired via a measurement matrix $A$ satisfying certain conditions (Candès & Tao, 2005; Donoho, 2006; Candès et al., 2006). Algorithmically, recovery is done via LASSO which solves for a convex optimization problem (Tibshirani, 1996; Bickel et al., 2009).

In statistical compressive sensing (Yu & Sapiro, 2011), we are given a finite set of training signals sampled i.i.d. from an unknown data distribution $p_{\text{data}}$ and the goal is to design an algorithm that minimizes the reconstruction error. Intuitively, there are two external factors to consider while designing a compressive sensing pipeline: the distribution of signals $p_{\text{data}}$ and the measurement noise $\epsilon$. The pipeline itself is a two step procedure, wherein the first step is the acquisition of the measurements for the signals under the measurement matrix $A$, and the second step is the efficient reconstruction of the signal by solving for an appropriate inverse optimization problem.

We propose Uncertainty Autoencoders (UAE), a framework that learns *both* the acquisition (a.k.a. encoding) and the recovery (a.k.a. decoding) procedures for compressive sensing given a fixed budget on the number of measurements (*i.e.*, $m$). For learning a UAE, we design an objective that maximizes for a variational lower bound on the empirical mutual information between $x^*$ and $y$. The encoders and decoders are parameterized via neural networks and optimized jointly in an end-to-end manner. Unlike the vast majority of prior work in statistical compressive sensing, our framework does not explicitly model strong assumptions such as sparsity (Baraniuk et al., 2010), or restricts the recovered signals to lie on a range determined by a generative model (Bora et al., 2017).

Our proposed algorithmic framework is both computationally and statistically efficient. By amortizing the recovery procedure, UAEs can scale to massive datasets, unlike prior frameworks that solve an optimization problem at test time for each new data signal. In terms of statistical performance, we demonstrate $20 - 46\%$ improvements in L2 reconstruction accuracies over competing approaches in the reconstruction errors for compressive sensing on the MNIST dataset.

## 2 LEARNING AND INFERENCE IN UNCERTAINTY AUTOENCODERS

We use upper-case symbols to denote probability distributions and assume they admit absolutely continuous densities on a suitable reference measure, denoted by the corresponding lower-case notation. Consider two multivariate random variables $X$ and $Y$ defined over $\mathbb{R}^n$ and $\mathbb{R}^m$ respectively. Our goal is to learn a forward mapping $f_\theta : \mathbb{R}^n \to \mathbb{R}^m$ from $X$ to $Y$ such that:

$$Y = f_\theta(X) + \epsilon \tag{2}$$

where we assume a suitable noise model for $\epsilon$, for *e.g.*, centered Gaussian noise with fixed variance. If $f_\theta$ is linear in $X$ (such as a neural network with no hidden layers), then we recover the system of equations in Eq. (1). In order to learn $\theta$ that permits easy acquisition and recovery of $x$, we propose to maximize the mutual information between $X$ and $Y$:

$$\max_\theta I_\theta(X, Y) = H(X) - H_\theta(X|Y) \tag{3}$$

where $H$ denotes the differential entropy.[1] Since the first term does not depend on $X$, we can equivalently minimize the conditional entropy $H_\theta(X|Y)$ as:

$$\min_\theta H_\theta(X|Y) = \mathbb{E}_{x^*, y \sim P_\theta(X,Y)}[\log p_\theta(x^*|y)]$$
$$= \mathbb{E}_{x^*, y \sim P_\theta(X,Y)}[\log p_\theta(x^*, y) - \log p_\theta(y)] \tag{4}$$

where $P_\theta(X, Y)$ is a joint distribution over the signals and the measurements. Moreover, we assume that the joint distribution factorizes, *i.e.*, $P_\theta(X, Y) = P(X)P_\theta(Y|X)$. The observation model $P_\theta(Y|X)$ depends on the noise model for $\epsilon$. A common choice is to assume a Gaussian noise model, and hence we have $P_\theta(Y|X) = \mathcal{N}(f_\theta(X), \sigma^2 I_m)$ for some positive scalar $\sigma$.

The data distribution $P(X)$ is however unknown and accessible only via a finite set of $D$ independent samples $\mathcal{X} = \{x_i^* \in \mathbb{R}^n\}_{i=1}^D$. Consequently, an estimate of the joint density $p_\theta(x^*, y)$ in Eq. (4) based on an empirical estimate of $P(X)$ can have high variance in practice. Hence, we introduce a factorized, variational approximation $Q_{\theta,\phi}(X, Y) = Q_\phi(X|Y)P_\theta(Y)$ parameterized by $\phi$. Substituting for the variational approximation, we get a lower bound to the objective in Eq. (4):

$$\mathbb{E}_{x^*, y \sim P_\theta(X,Y)}[\log p_\theta(x^*, y) - \log p_\theta(y)] \geq \mathbb{E}_{x^*, y \sim P_\theta(X,Y)}[\log q_{\theta,\phi}(x^*, y) - \log p_\theta(y)]$$
$$= \mathbb{E}_{x^*, y \sim P_\theta(X,Y)}[\log q_\phi(x^*|y)]$$

where the inequality follows from non-negativity of KL-divergence. Hence, our proposed framework optimizes for the following objective:

$$\min_{\theta,\phi} \frac{1}{D} \sum_{i=1}^D \mathbb{E}_{y \sim P_\theta(Y|X=x_i^*)}\left[\log q_\phi(x_i^*|y)\right]. \tag{5}$$

We parameterize both the encoder and decoder using feedforward neural networks and learn these parameters jointly via gradient methods. We refer to the above modeling and learning framework as Uncertainty Autoencoder (UAE) because it explicitly seeks to minimize the uncertainty in signal recovery from the measurements. If we assume no measurement noise (*i.e.*, $\epsilon = 0$) and assume the observation model $q_\phi(x_i^*|y)$ to be a Gaussian with mean $\mu_\phi(y)$ and a fixed variance $\sigma$, then the UAE objective reduces to minimizing the mean squared error between $x^*$ and $\mu_\phi(y)$. This special case of a UAE corresponds to a regular autoencoder (Bengio et al., 2009) where the measurements $Y$ signifying a hidden representation for $X$.

During test time, we directly observe the measurements $y_{\text{test}}$ that are assumed to be satisfying Eq. (2) for a target $x_{\text{test}}^*$. Hence, we can directly recover the signal by performing a forward pass through the decoder. For a Gaussian observation model, the mean and the mode coincide and hence the reconstruction $\hat{x}$ based on an MLE or MAP decoding is simply given as $\hat{x} = \mu_\phi(y_{\text{test}})$ with the L2 reconstruction error given by $\|x_{\text{test}}^* - \hat{x}\|_2$.

---

[1]Our analysis extends to the discrete setting as well, where one would consider the Shannon entropy.

Table 1: Average test L2 reconstruction errors (per image) for the MNIST dataset.

| Number of measurements (m) | 2 | 5 | 10 | 25 |
|:---:|:---:|:---:|:---:|:---:|
| Gaussian + LASSO | 8.81 | 8.85 | 8.85 | 8.89 |
| Gaussian + VAE | 8.95 | 8.06 | 7.09 | 3.95 |
| UAE | **7.06** | **4.39** | **3.80** | **2.78** |

## 3 RELATED WORK

The UAE framework is deceptively close to that of denoising autoencoders (DAE) and variational autoencoders (VAE). However, a DAE (Vincent et al., 2008) is different in the sense that it adds noise at the level of the input signal $X$, unlike a UAE where noise is assumed to be added at the level of the mapped signal $f_\theta(X)$. A VAE (Kingma & Welling, 2014)) on the other hand explicitly models the prior on the latent variables during decoding while the decoding phase of a UAE only models the likelihood $Q_\phi(X|Y)$.

The UAE objective, is closely related to the information maximizing objective, also referred to as InfoMax (Bell & Sejnowski, 1997). The InfoMax objective seeks to optimize for projections of the data that maximize the mutual information. A variant of the objective has been applied for compressive sensing (Weiss et al., 2007; Chang et al., 2009). However, these works considers a non-amortized setting and hence does not learn a decoder. Instead, it fits a *linear* measurement matrix to a per-data point variational approximation of the true posterior and is hence computationally expensive both during training and test time recovery.

## 4 EXPERIMENTS

The goal of our experiments is to evaluate the the effect of jointly learning the acquisition and the reconstruction procedures using uncertainty autoencoders. In order to do so, we performed compressive sensing on the MNIST dataset of handwritten digits (LeCun et al., 2010) with extremely low number of measurements. In particular we considered $m \in \{2, 5, 10, 25\}$. We assume a Gaussian noise model with $\sigma = 0.01$. Given a set of fixed budgets on the number of measurements, we evaluated the reconstruction error for the following benchmark measurement matrices and decoders.

- LASSO decoding with random Gaussian matrices. This decoder is based on a sparsity assumption on the underlying signal and solves for a convex L1-minimization problem such that $\hat{x} = \arg\min_x \|y - Ax\|_1$

- VAE decoding with random Gaussian matrices. Consider a trained variational autoencoder $\mathcal{P}(X, Z)$ over X and latent variables $Z \in \mathbb{R}^k$. Letting the mean function of the observation model $\mathcal{P}(X|Z)$ be denoted as $G : \mathbb{R}^k \to \mathbb{R}^n$, then this decoder solves for $\hat{x} = G(\arg\min_z \|y - AG(z)\|_2)$.

- UAE encoding and decoding. As discussed in Section 2, $\hat{x} = \mu_\phi(y)$.

The UAE encoder contains a hidden layer with 500 units. The decoder contains two hidden layers with 500 units each. We used ReLU activations and trained using Adam optimizer (Kingma & Ba, 2015) with a learning rate of 0.01 and a batch size of 100 trained over 100 epochs. The L2 reconstruction errors averaged over 100 test instances for all $m$ considered are shown in Table 1.

We observe that UAEs drastically outperforms both LASSO and VAE based decoding with random Gaussian matrices. In ongoing work, we are testing the UAE framework on more complex datasets. We are also excited about exploring connections of our learning objective with classic dimensionality reduction techniques such as principal component analysis and independent component analysis.

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
