# OpenReview forum: "Amortized Variational Compressive Sensing"
_ICLR.cc/2018/Workshop — Reject_

### Official Review · AnonReviewer3 · 2018-03-04
**Well-explained, and conceptually simple method, but motivations for the considered problem lacking**

**Rating:** 4
**Confidence:** 2

**Review:**

Summary
In traditional compressive sensing, a signal is reconstructed from linear measurements, using some a priori information about the signal (sparsity). In this article, the authors propose to replace the linear measurements by a learned encoder, and the reconstruction by a learned decoder, that does not explicitely use the a priori information. They derive a suitable objective function for the training, based on the notion of mutual information, and present experiments on MNIST.

Novelty: From a technical point of view, it seems to me that this article mostly reuses known techniques (notably coming from denoising as well as variational autoencoders). However, the authors apply these techniques to a different problem.
Clarity: very good.
Significance: unclear to me. As explained below, it probably depends which concrete goal the authors have in mind (which is not clear from the article).
Quality: very good; the mathematical derivations seem sound, and well-explained.

Cons
1. For me, the main issue is that the authors do not explain why they consider the problem of reconstructing a signal from a noisy representation of it given by a learned neural network. I think the motivations need to be seriously discussed.
It does not seem to me that the method proposed in this work can replace "traditional" compressed sensing. Indeed, in most applications of compressed sensing (notably MRI), as far as I know, the representation cannot be chosen at will: it is constrained by the physical devices that are used to compute it, although some parameters of it can possibly be freely chosen. In particular, in MRI, it has to be linear. So I do not think it makes sense to consider replacing it by a representation given by a neural network.
Another possibility is that the authors could be willing to use their method for signal compression. But in this area, I think there is already a significant literature, that I am not sure the method proposed in this article can compete with.
2. As stressed by the authors, when q_\phi is chosen to be Gaussian, the objective function (5) reduces to minimizing the mean square reconstruction error (and if q_phi is not Gaussian, then it amounts to minimizing some different reconstruction error). This is a natural objective, that could have been derived without using mutual information. So, in my opinion, it would be better to clarify what this "variational" setting, and the explicit modeling of the distribution of X as a function of Y, can bring for the problem at hand.

Pros
1. The method proposed by the authors is conceptually simple. Once trained, the algorithm is fast and, according to the numerical experiments, it works well. So if the authors have a precise application in mind, then their method is probably at least a very good benchmark.

Typos and minor remarks
- Abstract: "with much fewer measurements" -> "with much fewer measurements than ...".
- The term "amortize", in this context, is not clear to me; I think it would be better to succintly define it.
- Second line of page 2: "on a suitable reference measure" -> isn't it "with respect to ..."?
- After Equation (4): I do not understand "we assume that the joint distribution factorizes". All distributions satisfy the equality P_\th(X,Y) = P_\th(X) P_\th(Y|X). It would be better to rephrase or clarify this sentence.
- Is the variance sigma of P_\th(Y|X) assumed to be the same as the variance of q_\phi(x_i^*|y)? If yes, it would be better to clearly say it. Otherwise, the values of the two variances should be given in the "Numerical experiments" section.
- End of page 2: the sentence "where the measurements ... for X" is not clear to me.
- Page 3: there is one parenthesis too much after "(Kingma & Welling)".
- Page 3: no comma after "The UAE objective".
- First line of Section 4: "the the" -> "the".
- Section 4: "a set of fixed budgets" -> "a fixed budget".

---

### Official Review · AnonReviewer1 · 2018-03-10
**Interesting method**

**Rating:** 7
**Confidence:** 4

**Review:**

An interesting approach to compressed sensing structured signal, using a variational approach.

The method sounds promising, and certainly deserve exposition.

It is hard to judge the efficiency from these simulation though, but maybe this is unavoidable for such a short paper.

A side note: another approach for structured compressed sensing that bear similarities has been proposed in https://arxiv.org/abs/1606.03956 and https://arxiv.org/abs/1702.03260 where a Boltzmann machine (instead of a VAE as in the present work) learned to represent MNIST and then is used as a prior in the reconstruction. This also led to significant increases in performances and accuracy.

---

### Official Review · AnonReviewer2 · 2018-03-10
**The main novelty seems to be learning both an encoder and decoder.  Experimental work and comparisons is very limited.**

**Rating:** 4
**Confidence:** 5

**Review:**

There is a huge amount of work on compressed sensing, including many Bayesian approaches that use designed or learned priors.  The proposed approach appears to have some novelty, but the limited experiments and comparisons make it impossible to gauge the potential of the approach.

---

### Author Response · Authors · 2018-02-21
**Minor typos**

Minor typos the authors of this paper wished to highlight:

1. Below Eq. (3), we write the "first term does not depend on X" when it should have been "first term does not depend on \theta".

2. In Eq. (4), we should have:
max_\theta - H_\theta (X | Y)    [instead of min_\theta H_\theta (X | Y)]
Rest of the terms in the equation are fine.

3. In Eq. (5), we should have:
max_{\theta, \phi} 1/D...  [instead of min_{\theta, \phi} 1/D...]

We will fix these typos in a revision.

---

### Decision · Program_Chairs · 2018-03-20
**ICLR 2018 Workshop Acceptance Decision**

**Decision:**

Reject

**Comment:**

Based on the reviews, this paper has not been accepted for presentation at the ICLR workshop. However, the conversation and updates can continue to appear here on OpenReview.